# Prognostic Models in Growth-Hormone- and Prolactin-Secreting Pituitary Neuroendocrine Tumors: A Systematic Review

**DOI:** 10.3390/diagnostics13122118

**Published:** 2023-06-19

**Authors:** Roxana-Ioana Dumitriu-Stan, Iulia-Florentina Burcea, Teodor Salmen, Catalina Poiana

**Affiliations:** 1Department of Endocrinology, ‘Carol Davila’ University of Medicine and Pharmacy, 020021 Bucharest, Romania; iulia.burcea@umfcd.ro (I.-F.B.); endoparhon@gmail.com (C.P.); 2Doctoral School of ‘Carol Davila’ University of Medicine and Pharmacy, 050474 Bucharest, Romania; teodor.salmen@drd.umfcd.ro; 3‘C. I. Parhon’ National Institute of Endocrinology, 011863 Bucharest, Romania

**Keywords:** pituitary neuroendocrine tumor, GH-secreting pituitary neuroendocrine tumor, prolactinoma, prognostic factors

## Abstract

Growth-hormone (GH)- and prolactin (PRL)-secreting PitNETs (pituitary neuroendocrine tumors) are divided into multiple histological subtypes, which determine their clinical and biological variable behavior. Proliferation markers alone have a questionable degree of prediction, so we try to identify validated prognostic models as accurately as possible. (1) Background: The data available so far show that the use of staging and clinical–pathological classification of PitNETs, along with imaging, are useful in predicting the evolution of these tumors. So far, there is no consensus for certain markers that could predict tumor evolution. The application of the WHO (World Health Organisation) classification in practice needs to be further evaluated and validated. (2) Methods: We performed the CRD42023401959 protocol in Prospero with a systematic literature search in PubMed and Web of Science databases and included original full-text articles (randomized control trials and clinical trials) from the last 10 years, published in English, and the search used the following keywords: (i) pituitary adenoma AND (prognosis OR outcome OR prediction), (ii) growth hormone pituitary adenoma AND (prognosis OR outcome OR prediction), (iii) prolactin pituitary adenoma AND (prognosis OR outcome OR prediction); (iv) mammosomatotroph adenoma AND (prognosis OR outcome OR prediction). (3) Results: Two researchers extracted the articles of interest and if any disagreements occurred in the selection process, these were settled by a third reviewer. The articles were then assessed using the ROBIS bias assessment and 75 articles were included. (4) Conclusions: the clinical–pathological classification along with factors such as GH, IGF-1, prolactin levels both preoperatively and postoperatively offer valuable information.

## 1. Introduction

Pituitary neuroendocrine tumors (PitNETs) are benign monoclonal tumors that can cause symptoms due to their hormonal secretion or invasiveness. PitNETs are the third most common type of brain tumors (10–15% of all brain tumors) [1]. Despite their benign evolution, in some cases, they can have aggressive behavior. Based on these observations came the proposal of the European Pituitary Pathology Group (EPPG) to replace the term ‘adenoma’ with that of neuroendocrine pituitary tumor, which emphasizes the fact that these tumors may demonstrate unpredictable evolution [2]. In addition, the important role of pathology in the prediction of the outcomes has been stated, using histology associated with immunohistochemistry (IHC) and the expression of transcription factors, along with the Ki-67 labeling index, p53, and O6-methylguanine-DNA methyltransferase (MGMT) expression. However, there are several different opinions among researchers about the methods used to evaluate these markers, including immunohistochemistry versus real-time PCR [2,3].

Another important clinicopathological classification is the five-tiered prognostic classification of PitNETs proposed by a French group in 2013 [3], which was validated on a total of 2206 patients [3,4,5,6,7,8]. 

In 2022, the updated WHO classification recognizes the importance of other immunohistochemical markers, besides pituitary transcription factors and hormone immunohistochemistry. In addition, the term ‘atypical adenoma’ is no longer recommended [9]. 

Endocrine pituitary tumors arise from adenohypophyseal cells and they are traditionally classified based on hormonal activity as non-functional or functional, but the latest WHO classifications emphasize the importance of cell-lineage-based classification, characterized by lineage-specific transcription factors (three or more lineages: lactotroph, somatotroph, somatotroph, and thyrotroph belong to PIT-1—pituitary-specific TF 1, corticotroph, which belongs to T-PIT pituitary cell-restricted factor and gonadotroph, which is related to SF-1—steroidogenic factor 1), and tumors that are negative for all TFs are considered ‘null cell’ adenomas [9]. 

The classification of PitNETs has evolved over the years with continuous improvements based on novel technologies. Experts propose a multistep approach in these tumors that should comprise the following: clinical data, radiological characteristics (new magnetic resonance imaging (MRI) techniques and some studies also propose radiomic evaluation of consistency that can have prognostic value), histological reports, immunohistochemistry analysis for hormones, cytokeratin low-molecular-weight keratin patterns (LMWK), the ki-67 labeling index, somatostatin receptors (SSTRs), estrogen receptor alpha (ERα), and methylation of O6-methylguanine-methyltransferase (MGMT) [2,3]. In addition, machine learning (ML) and artificial intelligence (AI) are advancing and have led to new methods to more accurately predict the evolution and prognosis of these tumors. 

## 2. Materials and Methods

We registered a systematic review protocol under the number CRD42023401959 in Prospero that followed the recommendations of Preferred Reporting Items for Systematic Reviews and Meta-Analyses (PRISMA) [10]. Furthermore, we used the Population, Intervention, Comparison, Outcome, and Study Design (PICOS) strategy to guide our study rationale and to conduct a clear, useful, and systematic literature search. We conducted a systematic literature search in the PubMed and Web of Science databases, which include original full-text articles, randomized control trials, and clinical trials, from 1 January 2013 up to 31 December 2022, published in English, with the following keywords: (i) pituitary adenoma AND (prognosis OR outcome OR prediction), (ii) growth hormone pituitary adenoma AND (prognosis OR outcome OR prediction), (iii) prolactin pituitary adenoma AND (prognosis OR outcome OR prediction), (iv) mammosomatotroph adenoma AND (prognosis OR outcome OR prediction). Two researchers extracted the articles of interest and they were then assessed using the ROBIS risk of bias recommendations [11]. We identified 114 articles in PubMed and 2173 articles in the Web of Science database. We preferred to use the search term ‘adenoma’ based on the relatively recent changes in terms. The exclusion criteria were duplicates, articles that lack originality, published in languages other than English, and on non-human populations. Two researchers, RDS and TS, extracted the included studies’ titles and abstracts, screened them for relevance for the present study theme and selected the relevant ones by performing cross-screening; if any disagreements occurred in the selection process, these were settled by a third reviewer. As a result, after the bias assessment using the ROBIS risk of bias recommendations used for systematic reviews [12], 75 articles were included, as shown in Figure 1. 

We also performed a manual search of the databases to identify other potentially useful articles missed by our search strategy. Finally, in the Discussion section, we completed this information to provide a more comprehensive picture of the available data from the literature. The relevance of each study selected was assessed by determining if it responded to the ‘target questions’. The review was conducted by two authors (RIDS and TS) and the goal was to include as many studies as possible. Excluded articles were case reports, reviews, meta-analyses, letters to the editors, articles not in English, abstracts or research that did not respect the goals of our review.

Data were extracted with regard to the following aspects:Study parameters: authors, title, year, design, and number of patients.Clinical parameters: GH and PRL secretion, tumor volume and surgical treatment.Imaging parameters: MRI evaluation and invasion.Histopathological, immunohistochemical and molecular analysis.Surgical outcome and postoperative complications.Morbidity and mortality.

## 3. Results

The results of our research are synthesized in Table 1, which contains data about the authors (in the “Author” column), date of publishing (in the “Date of publishing” column), the type of study (in “Study type” column), the total number of patients enrolled (in “Number of patients” column) and the main outcomes reported by each article in the results section (in “Main results” column).

Table 1 synthesizes the main results of studies that included patients diagnosed with GH-secreting PitNETs.

Table 2 synthesizes the main results of studies that included patients diagnosed with PRL-secreting PitNETs. 

Table 3 synthesizes the main results of the studies that included patients diagnosed with mixed PRL and GH secretion. 

Table 4 summarizes the studies that proposed models to predict endocrinological outcomes after pituitary surgery.

## 4. Discussion

### 4.1. Biochemical Factors

Cavernous sinus invasion, larger tumor size, higher preoperative GH levels and maximal tumor diameter are consistently associated with lower surgical remission rates in the literature [36].

Moreover, patients with smaller tumors and without CS invasion were more likely to be associated with surgical remission. The strongest predictors of remission are as follows: preoperative GH levels, cavernous sinus invasion, and the surgeon’s experience [37].

In the case of PRL-secreting PitNETs, the preoperative level of PRL is the only significant predictor of remission.

### 4.2. Radiological Parameters

MRI is the investigation of choice for the evaluation of PitNETs. Various parameters regarding the extent, consistency, and contrast uptake can be studied and a non-invasive diagnosis is possible. 

Based on the MRI characteristics, the Zurich Pituitary Score (ZPS) was created for PitNETs and in some cases, CT can also be used. It is a good predictor for postoperative residual tumor volume and gross total resection rate [37]. It has the following four grades: grade I: ZPS ratio ≤ 0.75; grade II: 0.75 < ratio ≤ 1.25; grade III: 1.25 < ratio; grade IV: encasement of the intracavernous ICA (internal carotid artery, uni- or bilaterally). Intraoperative high-field MRI (3T-iMRI) is particularly advantageous when associated with ZPS and could be useful in some cases [38]. Currently there are few studies that show the utility of intraoperative 3T-iMRI. 

Tumor consistency has been found as a predictor for surgical outcomes. Preoperative evaluation of consistency is based on MRI techniques (7 T MRI); tumors rated as ‘soft’ intraoperatively were hyperintense compared to local gray matter on T2-weighted imaging and ‘soft’ tumors presented with a higher vascularity than ‘firm’ tumors [38,39].

### 4.3. Invasion and Proliferation

Based on the five-tiered classification, the invasion of PitNETs is defined as histological and or radiological MRI signs of cavernous or sphenoid sinus invasion [5]. 

Proliferation is considered in the presence of at least two of three markers, which are as follows: mitoses count: n > 2/10 HPF; ki-67 index ≥ 3%; p53 detection positivity >10 strongly positive nuclei/10 HPF [5]. 

Knosp grade, Knosp modification and Hardy–Wilson classifications are widely used and offer a good orientation. The Knosp grade offers the greatest diagnostic accuracy for the prediction of a surgical cure [40,41,42]. In addition, the degrees of ICA contact and Knosp score are associated with postoperative endocrine outcomes [43].

### 4.4. Pathological and Molecular Factors

Multiple predictive variables (age, gender, preoperative GH and IGF-1 levels, maximal tumor diameter, Hardy’s and Knosp’s grade, MRI T2-weighted tumor intensity, cytokeratin expression pattern, and clinicopathological classification) are related to postoperative outcomes. Younger age, higher preoperative GH and- or IGF-1 levels, group 2b of the clinicopathological classification, Knosp’s grade IV, MRI, T2-weighted tumor hyperintensity and sparsely granulated cytokeratin expression patterns are related to worse postoperative outcomes [44,45,46]. Molecular, pathological, biochemical, radiological and individual factors are resumed in Figure 2.

### 4.5. Age and Sex

Younger age was associated with a low probability of achieving long-term IGF-1 normalization; in addition, the preoperative growth rate is associated with age, residual tumor volume with older age, gender and suprasellar CS extension [33,47,48,49]. However, age is not likely to be a predictor of surgical outcomes in general [48,49,50,51,52].

In the case of PRL-secreting PitNETs, it is well known that male patients have larger tumors and are more likely to be resistant to treatment [52,53,54].

### 4.6. Clinical Prediction Models

Current clinical practice is mainly concentrated on establishing the diagnosis, etiology and choosing the best therapeutic approach. Predicting the evolution of a certain disease can be challenging, as one or multiple variables may be needed, and a multi-step and multivariable/multifactorial approach is mandatory for the design and analysis of a certain prediction model. Models are needed, as clinicians try nowadays to personalize each treatment that they prescribe, for a better outcome [54,55,56].

The Prognosis Research Strategy (PROGRESS) group has proposed a number of methods to improve the quality and impact of model development [2,16]. Furthermore, investigators into the transparent reporting of a multivariable prediction model for the Individual Prognosis Or Diagnosis (TRIPOD) study have established a checklist of recommendations for reporting on prediction or prognostic models [17,57,58]. 

The five steps that must be checked before creating a prognostic model are firstly the research question (in our case: what are the variables that predict the cure of patients with GH- and PRL-secreting PiTNETs?), dataset selection, variables (generally including more than 10 variables may decrease efficiency and researchers also state that variables that are not statistically significant may contribute to the model), model generation (regression analyses, including linear, logistic, and Cox models are widely used depending on the model and its intended purpose and to check for overfitting of the model), Akaike information criterion (AIC) [28] and an index of model fitting that charges a penalty against larger models, which may be useful [20]. Lower AIC values indicate a better model fit. Some interpret that the AIC addresses the explanation aspect and Bayesian information criterion (BIC) addresses prediction, where BIC may be considered a Bayesian counterpart, and the last step of model validation is external/internal [58,59].

Until now, for PitNETs, the clinicopathological five-tiered classification proposed by Trouillas et al. to predict the evolution of pituitary adenomas has been one of the most reliable methods for predicting the evolution of these patients. The classification has been externally validated on large cohorts, including a total of 2206 patients in the following studies: retrospective multicentric analysis of 410 patients with a postoperative follow-up period of 8 years (2013, Trouillas et al.); analyses of disease-free and recurrence/progression-free status revealed the significant prognostic value (*p* < 0.001; *p* < 0.05) of age, tumor type, and grade across all tumor types and for each tumor type. Invasive and proliferative tumors (grade 2b) had a poor prognosis with an increased probability of tumor persistence or progression of 25- or 12-fold, respectively, as compared to non-invasive tumors (grade 1a) [3]; Raverot G et al. in 2017 conducted a prospective study with 374 patients included, which were retrospectively evaluated, and showed that the tumor grade was a significant predictor of progression-free survival (*p* < 0.001). A multivariate analysis indicated grade (*p* < 0.001), age (*p* = 0.035), and tumor type (*p* = 0.028) as independent predictors of recurrence and/or progression. This risk was 3.72-fold higher for a grade 2b tumor compared with a grade 1a tumor [5]; Lelotte et al. in 2018 retrospectively analyzed a cohort of 120 patients with a follow-up of 48 months and showed that the risk of recurrence/progression was 8.6-fold higher for grade 2b tumors as compared with grade 1a tumors, age, the presence of a postoperative residual tumor (*p* < 0.001) and the proliferative nature of the tumor were all independent factors predicting recurrence or progression [6]; Asioli S et al. in 2019 carried out a retrospective analysis of 566 patients with pituitary adenomas, specifically 253 FSH/LH-secreting, 147 GH-secreting, 85 PRL-secreting, 72 ACTH-secreting and 9 TSH-secreting tumors, with at least 3-year post-surgical follow-ups that validated the score and confirmed multivariate Cox regression analysis showed tumor invasion (HR = 1.926; *p* = 0.001), Ki-67 ≥3% (HR = 2.290; *p* = 0.003) and type to be an independent prognostic factor of disease-free-survival [7].

The five-tiered classification was also validated in a study in 2022 (Sahakian N et al.) on a cohort of 607 patients with a median follow-up period of 38 months, which confirmed once more that tumor grades were significant and independent predictors of PFS (*p* < 0.001) with a 4.8-fold higher risk of progression/recurrence in grade 2b tumors as compared to grade 1a tumors. In addition, in this study, proliferative tumors exposed the patient to a 9.5-fold higher risk of having ≥3 adjuvant therapeutic lines as compared to non-proliferative tumors [8]. Another study published in April 2023 (Peixe. C. et al.) that analyzed a cohort of 129 patients showed once again that grade 2b tumors and significantly higher rates of persistent tumor remnants within 1 year after the operation (93-78-18-30%; *p* < 0.001), active disease at last follow-up (40-27-12-10%; *p* = 0.004), re-operation (27-16-0-5%; *p* = 0.023), irradiation (53-38-12-7%; *p* < 0.001), multimodal treatment (67-49-18-25%; *p* = 0.003) and multiple treatment (33-27-6-9%; *p* = 0.017) were important factors [50]. Patients with grade 2b Pas also required a higher mean number of treatments (2.6-2.1-1.2-1.4; *p* < 0.001).

### 4.7. Artificial Intelligence and Machine Learning for Prognosis

Technology is continuously evolving and in the precision medicine era, we need new methods that can help clinicians to establish a complex and accurate diagnosis and to identify certain factors that may indicate resistant or aggressive behavior [60,61,62]. 

AI is a methodology of computer systems that uses algorithms to tirelessly process data, automatically learn and understand its meaning, generate computer models, and identify the best predictive features present in training data [3]. As a domain of AI, ML is defined as performing automated learning from the input or data (experience) that it has been presented, and it converts these data to expertise or knowledge. ML can be used to design and train software algorithms to learn from and act on data [63,64,65,66,67,68,69].

The preoperative variables included age, gender, BMI, treatment history (surgery, medication and radiotherapy), MRI features (tumor dimensions, Knosp classification and clivus invasiveness), serum random GH and serum IGF-1 levels. The intraoperative variables included the following: surgeons’ experience (based on annual pituitary operations performed), operative approach, total resection or subtotal (based on operation note), intraoperative cavernous sinus invasion, tumor texture, CSF leakage, and the presence of pseudocapsules. The only postoperative variable was the serum GH level in the first postoperative morning. The partial model only included preoperative variables, and the full model integrated all the variables. Several algorithms (penalized logistic regression, gradient boost machine, support vector machine, neural network, and an ensemble algorithm) were used to train the partial model and the full model, respectively. The internal performance was assessed by 10-fold cross-validation. Following this study, an online application was launched—acro-remission prediction. Feature importance explanations showed that the Knosp grade was the most important variable for the partial model and the postoperative day 1 GH level, total resection, and Knosp grade were the three most important variables for the full model. Gross total resection was correlated in many studies with remission after transsphenoidal surgery, and the patients were more likely to have favorable endocrine and visual outcomes than those with incomplete resection [70,71,72,73]. 

Radiomics has emerged in the field of medical imaging analysis in recent years [74,75,76,77]. This method has been used in the diagnosis or prognosis of colorectal cancer, non-small-cell lung cancer, and gliomas [77,78,79,80]. In the case of PitNETs, several studies have shown that this non-invasive radiologic method can preoperatively predict CS invasion using the Knosp grade. After image acquisition (contrast-enhanced T1- and T2-weighted MR imaging), feature extraction and analysis, models were created using a support vector machine [81,82,83]. The clinico-radiological risk factors comprised gender, age, tumor volume, Knosp grade (2 or 3), tumor diameter, hemorrhage (yes or no), suprasellar invasion (yes or no), periarterial enhancement (yes or no), and inferolateral venous compartment obliteration (yes or no). After statistical analysis, a nomogram was created to validate the test set [84,85,86].

Our personal approach in clinical practice includes the mandatory evaluation of biochemical, imaging, histopathological and immunohistochemical evaluation after surgery. Based on our experience, immunohistochemistry is very important in helping clinicians to organize personalized therapy. Rare cases of plurihormonal PitNETs are examples that prove the usefulness of clinicopathological classification and molecular biomarkers in personalized treatment.

### 4.8. Strengths and Limitations

The major strength of our review is the fact that it is the most recent review of the literature concerning the factors that are associated with the outcome of GH and PRL PitNETs. We included the most relevant and most recent data available at this moment. Furthermore, our work provides future perspectives on AI and ML techniques that can create relevant prognostic models, which can help clinicians to apply personalized treatment. At present, there is no consensus regarding what clinical, imaging, pathological or molecular factors should be mandatory during evaluation following diagnosis.

The primary limitation of our study is the large number of clinical trials available that frequently included a small number of patients and evaluated different types of parameters. IN addition, there is a difference regarding the laboratory methods used to evaluate hormonal status and some studies had a lack of data regarding immunohistochemical, histopathological and molecular evaluation. 

## 5. Conclusions

Our review of the currently available data in the literature confirms that age, gender, tumor volume, radiological characteristics, invasion, proliferation, preoperative and postoperative biochemical parameters and the neurosurgical experience are factors that can predict the surgical outcome of GH- and PRL-secreting PitNETs. Integrating these parameters and external validation into large cohorts of certain prognostic models is needed; ML techniques and AI represent future directions that should be investigated. Clinical practice showed that the anterior classifications and diagnostic methods used in the case of these tumors are insufficient. The use of immunohistochemistry, molecular markers and complex pre-operative radiological examinations that improve surgical precision and support total resection (radiomics, high-resolution MRI and ultra-high field 7 T MRI) can assure a high cure rate.

## Figures and Tables

**Figure 1 diagnostics-13-02118-f001:**
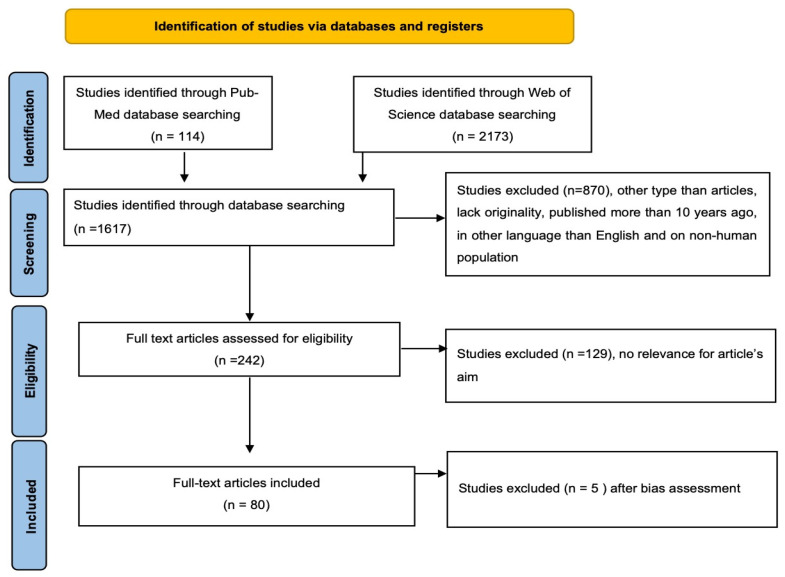
Flowchart of the study selection process according to PRISMA guidelines.

**Figure 2 diagnostics-13-02118-f002:**
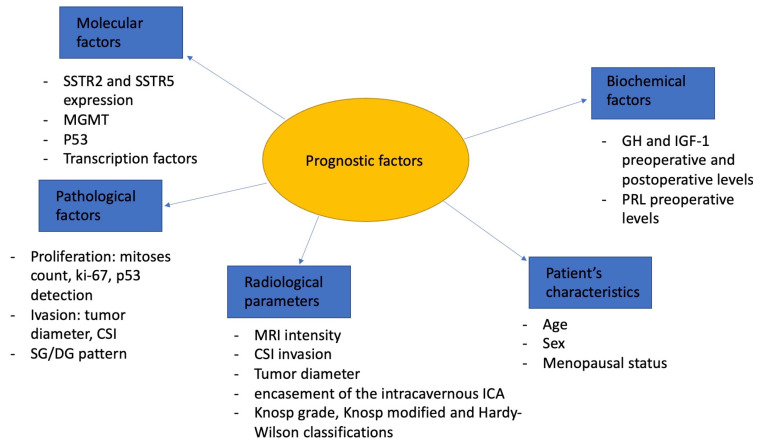
Main prognostic factors in GH- and PRL-secreting PitNETs. SSTR2—somatostatin type 2 receptor; SSTR5—somatostatin type 5 receptor; MGMT—O(6)-methylguanine-DNA methyltransferase; CSI—cavernous sinus invasion; SG—sparsely granulated; DG—densely granulated; MRI—magnetic resonance imaging; ICA—internal carotid artery; GH—growth hormone; PRL—prolactin.

**Table 1 diagnostics-13-02118-t001:** Summary of the results: prognostic factors and outcomes for GH-secreting PiTNETs.

Author	Date of Publishing	No. of Patients	Study Type	Main Results
Alhambra-Expósito et al. [12]	2018	22	Retrospective Single Center	Hyperintensity on MRI correlates with invasion in the CS and extrasellar growth (*p* = 0.023)
Donegan D. et al. [13]	2022	106	Retrospective Single Center	Early postoperative IGF-1 concentration, index or % change from diagnosis (at 6 weeks) were the best predictors of surgical outcome (*p* < 0.001)
Goyal-Honavar A et al. [14]	2021	203	Retrospective Single Center	Preoperative basal GH value < 40 ng/mL (*p* < 0.001) and tumor size correlate with better rates of surgical remission
Heng L. et al. [15]	2021	83	Retrospective Single Center	SG subtype presented with tumors significantly larger compared with the DG group (*p* < 0.001) and had a higher Knosp grade (*p* = 0.0012) The grading scale for predicting SG GH-secreting pituitary adenomas with AUC = 0.84 (DKGO score)
Liu X. et al. [16]	2022	44	Retrospective Single Center	Ki-67 index was higher in the refractory tumors group (mean 8.6%) vs. refractory tumors group (*p* < 0.001); EGFR increased in refractory tumors (*p* < 0.01)
Ozturk et al. [17]	2020	67	Retrospective Single Center	Pre-operative IGF-1 correlates with ESR1 Ct (r = 0.373, *p* = 0.30) and ESR2 Ct (r = 0.48, *p* = 0.017). High ER expression in acromegalic patients associated with a decrease in pre-operative IGF-1 only in male patients
Park HH et al. [18]	2018	132	Retrospective Single Center	Tumors with far-lateral TSA showed significant reductions in postoperative nadir GH at 1 week (*p* = 0.014), 6 months (*p* < 0.01), and 1 year (*p* = 0.018), and in IGF-I at 1 year (*p* < 0.01)
Park SH et al. [19]	2017	463	Retrospective Single Center	Premenopausal females had a higher proportion CSI compared to males aged < 50 years (35.3% vs. 21.7%, *p* = 0.007) and had significantly lower long-term remission rates than males aged < 50 years (69.3% vs. 88.0%, *p* < 0.001)
Swanson et al. [20]	2021	131	Retrospective Single Center	SG-As were larger (*p* = 0.03), more frequently invasive (*p* = 0.004), associated with higher MRI, T2-weighted signal ratio (*p* = 0.01), and showed lower SSTR2 expression (*p* < 0.0001)
Ferrés et al. [21]	2023	44	Retrospective Single Center	SG was related to a reduced probability for IGF-1 normalization (*p* = 0.01), augmented recurrence risk (RR = 34.5, *p* = 0.01), and a significant need for reintervention (*p* = 0.014)
Coopmans EC. et al. [22]	2021	282	Retrospective Multicenter	Higher random GH concentration at diagnosis and a larger maximum tumor diameter associated with a lower change in long-term remission (OR = 0.93, 95% CI 0.89–0.97, *p* = 0.0022, respectively; OR = 0.98, 95% CI 0.96–0.99, *p* = 0.0053)

GH—growth hormone; MRI—magnetic resonance imaging; CSI—cavernous sinus invasion; SG—sparsely granulated; DG—densely granulated; TGR—tumor growth rate; EGFR—Epidermal Growth Factor Receptor; ESR1—Estrogen Receptor 1; ESR2—Estrogen Receptor 2; Ct -cycle threshold; TSA—microsurgical transsphenoidal approach; SSTR2—somatostatin receptor type 2; RR—relative risk.

**Table 2 diagnostics-13-02118-t002:** Summary of the results: prognostic factors and outcomes for PRL-secreting PiTNETs.

Author	Year of Publishing	No. of Patients	Study Type	Relevant Findings
Baussart, B. et al. [23]	2021	114	Retrospective Single Center Microadenomas	Preoperative PRL level is the only significant predictor of remission (*p* = 0.014) CSI was a predictor of lower remission (*p* = 0.009).
Vermeulen E. et al. [24]	2020	69	Retrospective Single Center	The 4 most powerful predictors are sex, tumor volume, the moment of prolactin normalization and the presence of a cystic, hemorrhagic, or necrotic component
Cander S. et al. [25]	2014	113	Retrospective Single Center	The rate of invasive pituitary adenoma is significantly higher in male patients
Han YL. et al. [26]	2018	52	Retrospective Single Center	Tumor size *p* = 0.007; OR = 5.748, 95% CI 1.621–20.379; and preoperative PRL levels *p* = 0.006; OR = 3.886, 95% CI 1.464–10.212 associated with unsatisfactory postoperative outcomes
Lv L. et al. [27]	2019	42	Retrospective Single Center Giant Prolactinomas	Male patients had larger tumors (14.57 vs. 7.74 cm^3^, *p* = 0.0179)

GH—growth hormone; PRL—prolactin; CSI—cavernous sinus invasion; OD—odds ratio; CI—confidence interval.

**Table 3 diagnostics-13-02118-t003:** Summary of the results: prognostic factors and outcomes for mamosommatotroph PitNETs.

Author	Year of Publishing	No. of Patients	Study Type	Relevant Findings
LV L et al. [28]	2019	94	Retrospective	Female gender was associated with recurrence (*p* = 0.0003) Tumor volume and gender predict long-term remission (*p* < 0.0001) CSI, suprasellar/sellar extension, tumor volume independent predictors for long-term biological remission
Monsalves et al. [29]	2014	153	Retrospective	Preoperative growth rate associated with: -age (*p* = 0.0001), -FGFR-4 (*p* = 0.047) -p27 negativity (*p* = 0.007). Residual tumor volume associated with older age (*p* = 0.038), gender and suprasellar CS extension.
Nikitin PV et al. [30]	2018	50	Retrospective	Ki-67 cytocolorimetric index was correlated with tumor relapse (*p* = 0.0027 × 10^–6^, z = 5.3)
Pappy et al. [31]	2019	501	Retrospective	Best prediction model (M1): -tumor diameter > 2.9 cm -CSI -ki-67 > 3%
Chen Y et al. [32]	2021	172	Retrospective	Knosp classification grade 4, partial resection and ≥3% Ki-67 positive rate are independent risk factors of tumor recurrence or progression (*p* < 0.05)

CSI—cavernous sinus invasion; CS—cavernous sinus; FGFR-4—Fibroblast Growth Factor Receptor 4.

**Table 4 diagnostics-13-02118-t004:** Summary of studies that proposed models to predict endocrinological outcomes after pituitary surgery.

Author	Year of Publishing	No. of Patients	Study Type	Relevant Findings
Qiao N et al. [33]	2021	883	Retrospective and Prospective	Knosp grade was the most important variable for the partial model. Postoperative day 1 GH level, total resection, and Knosp grade were the three most important variables for the full model.
Huber M. et al. [34]	2022	86	Prospective	Machine learning methods predict the outcome of surgery in prolactinomas. Super learner exhibits very good prediction for the primary outcome (mean AUC = 0.9, 95% CI 0.92–1.00).
Niu J. et al. [35]	2019	194	Retrospective	Nomogram using radiomics methods based on contrast-enhanced T1- and T2-weighted magnetic resonance predicted results better than the clinic-pathological model (AUC = 0.871, 95% CI 0.857–0.881 in test set, *p* = 0.021).
Zhang Y, Tu S et al. [36]	2021	38	Retrospective	The PPV of CS invasion of three-grade classifications is based on 3D MMI.

GH—growth hormone; AUC—area under the curve; CI—confidence interval; PPV—positive predictive value; CS—cavernous sinus; 3D-MMI—3D-multimode interference.

## Data Availability

Not applicable.

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
