# Peer review of "Prognostic Models in Growth-Hormone- and Prolactin-Secreting Pituitary Neuroendocrine Tumors: A Systematic Review"

_diagnostics, 2023, doi:10.3390/diagnostics13122118_

Round 1

Reviewer 1 Report

The title (“Prognostic Models in Growth-Hormone and Prolactin secreting Pituitary Neuroendocrine: a Systematic Review”) not indicates the study because of it needs of the word tumor (line 3).  

The abstract is well explained and summarized but the conclusions (lines 28 –29) must be reconsider; please improve the sentence “[...] can bring valuable information.”.   

The paper is clear, comprehensive and relevant for the field.  

The manuscript is well structured and the cited references contains recent publications. 

Tables and schemes are appropriate and easy to understand. 

Methods and results are well explained. 

The final conclusions are consistent.

Perhaps it is a distracting error but there isn’t the word "tumor" in the manuscript title (line 3).

Author Response

Thank you so much for the review and suggestions! I will make the changes and submit the revised version of the manuscript.

Best wishes,

Dr. Roxana Dumitriu.

Reviewer 2 Report

Comments to the authors. 

1. The flow chart. The numbers do not add up. E.g. 1617 patients and you extract 870 patients - and then you end with 242? After 242 you exclude 129 and end up with 80??

2. There are much data in the paper, I miss a meta-analysis in order to get a better focus on the results of the review. It is unclear at some points due to the description of many, many papers. Alternatively, you could focus your description of studies. Do you need to mention so many?

3. 

Please look at the "Results paragraph" - it needs to be re-written. 

Author Response

Thank you for the comments and suggestions, I have revised the manuscript based on the Review Report. I added some figures and revised the discussions and conclusions. 

Best wishes, 

Dr. Roxana Dumitriu

Reviewer 3 Report

General Comments

The initiative for this elaborate work is appreciated. A difficult task given the heterogeneity of the published data.

However, the description of the many small retrospective studies are tedious to read.

Unfortunately, there are many inaccuracies in the paper, which makes it even more difficult to read and understand, what the authors intend to convey.

Elaborate and thought-out  tables and flowcharts with weighting of the results would have been more appropriate and supportive for daily clinical practice and could have condensed the MS to the essential points.

Questions and remarks

1.  Title: 

Prognostic Models in Growth-Hormone and Prolactin secreting 2 Pituitary Neuroendocrine: a Systematic Review

should it not rather read:

Prognostic Models in Growth-Hormone and Prolactin secreting 2 Pituitary Neuroendocrine Tumors: a Systematic Review 

2. Abstract: there is no paragraph (3) "results"

Conclusions: The clinico-pathological classification along with factors like GH, IGF-1, prolactin levels preoperatively and postoperatively cand bring valuable information. 

"can" or "can't" ?

3. numbers of references are misplaced

4. line 103: "We also performed a manual search of the references to identify other potentially useful articles missed by our search strategy."

Do you mean the references of the papers retrieved ? If so, please clarify.

5. Tables 1-3 have only one legend, which is placed after Table 1, which is a bit confusing, since the other tables have not legend. Some abbreviations are missing e.g. CSI, M1, OR

Title of Table 1 is placed after the legend. Any reason ?

Table 2. Baussart B et al. Please explain  "CSI is a predictor of remission"

do you mean: a predictor for lower remission ?

6. Results and Discussion sections are not well separated

Please discuss and weight your personal expert opinion on how you proceed in daily practice after the discussion of the results.

Line 143: do you mean "oGTT" ?

Line 164 and line 171: please remove duplicates

Line 185: please write out Sen and Spe

Line 312-314: this sentence is quite difficult to understand

English language has to be improved

please see also comments to the authors

Author Response

Thank you for the comments and suggestions, I have revised the manuscript based on the Review Report. 

Best wishes, 

Dr. Roxana Dumitriu

Round 2

Reviewer 2 Report

Thank you for the revision, I have no further comments. 

Author Response

Thank you for the review! Best regards, Dr. Roxana Dumitriu-Stan.

Reviewer 3 Report

Although I appreciate the great work of literature search and analysis, I still think that the paper is tedious to read and the conclusions are vague.

Even after the revision by the authors, there are still too many grammar and typing errors (e.g. missing verb, incomplete sentences, misspellings, lost reference) in the manuscript.

Even after the revision by the authors, there are still too many grammar and typing errors (e.g. missing verb, incomplete sentences, misspellings, lost reference) in the manuscript.

Author Response

Dear Reviewer, 

Thank you for the comments and suggestions. Based on the observation that the conclusions are vague we have modified them trying to resume all the information extracted from the literature.

The second point - "there are still too many grammar and typing errors (e.g. missing verb, incomplete sentences, misspellings, lost reference) in the manuscript." - we checked once again the grammar and we send the manuscript for editing to a collegue, a native speaker. All the changes made are now highlighted with track changes. 

The third point - references missplaced -we checked again and we corrected all the references and removed the inappropriate ones based on the changes we made in the Results and Discussion Sections (we removed duplicates at the anterior revision).

Best regards,

Dr. Roxana Dumitriu-Stan
